# Molecular mechanism of setron-mediated inhibition of full-length 5-HT$_{3A}$ receptor

Sandip Basak [1,5], Yvonne Gicheru[1,5], Abhijeet Kapoor[2], Megan L. Mayer[3], Marta Filizola [2] & Sudha Chakrapani[1,4]

Serotonin receptor (5-HT$_{3A}$R) is the most common therapeutic target to manage the nausea and vomiting during cancer therapies and in the treatment of irritable bowel syndrome. Setrons, a class of competitive antagonists, cause functional inhibition of 5-HT$_{3A}$R in the gastrointestinal tract and brainstem, acting as effective anti-emetic agents. Despite their prevalent use, the molecular mechanisms underlying setron binding and inhibition of 5-HT$_{3A}$R are not fully understood. Here, we present the structure of granisetron-bound full-length 5-HT$_{3A}$R solved by single-particle cryo-electron microscopy to 2.92 Å resolution. The reconstruction reveals the orientation of granisetron in the orthosteric site with unambiguous density for interacting sidechains. Molecular dynamics simulations and electrophysiology confirm the granisetron binding orientation and the residues central for ligand recognition. Comparison of granisetron-bound 5-HT$_{3A}$R with the apo and serotonin-bound structures, reveals key insights into the mechanism underlying 5-HT$_{3A}$R inhibition.

[1] Department of Physiology and Biophysics, Case Western Reserve University, Cleveland, OH 44106-4970, USA. [2] Department of Pharmacological Sciences, Icahn School of Medicine at Mount Sinai, New York, NY, USA. [3] Division of CryoEM and Bioimaging, SSRL, SLAC National Accelerator Laboratory, Stanford University, Menlo Park, CA 94025, USA. [4] Department of Neuroscience, School of Medicine, Case Western Reserve University, Cleveland, OH 44106-4970, USA. [5] These authors contributed equally: Sandip Basak, Yvonne Gicheru. Correspondence and requests for materials should be addressed to S.C. (email: Sudha.chakrapani@case.edu)

Nausea and vomiting remain the most prevalent and debilitating side effects of cancer treatment by chemotherapy and radiation therapy. These symptoms drastically affect patient health and quality of life, and in turn impact chemotherapy compliance[1]. Radiation and cytotoxic drugs trigger the release of large amounts of serotonin from the mucosal enterochromaffin cells, which bind serotonin receptors (5-HT$_3$Rs) both on the vagal afferent nerve in the gut and on the chemoreceptor trigger zone in the brainstem, leading to the pathogenesis of emesis in these patients. 5-HT$_3$Rs belong to the pentameric ligand-gated ion channel (pLGIC) superfamily[2] and they mediate excitatory postsynaptic signaling. 5-HT$_3$Rs are expressed either as a homopentamer consisting of subunit A or as heteropentamers of subunit A in combination with other subunits (B, C, D, or E). 5-HT$_3$Rs are widely expressed in both the central and peripheral nervous system, and form a part of the brain–gut circuitry that directly control peristalsis, emetic reflex, and visceral pain perception[3–8]. Dysfunction in 5-HT$_3$Rs is implicated in several psychiatric and gastrointestinal conditions, such as bipolar disorder, anxiety, depression, and irritable bowel syndrome[6,9,10]. The 5-HT$_3$R antagonists, referred to as "setrons," are the standard first line of therapy for the prevention of chemotherapy-induced emesis. Additionally, 5-HT$_3$R inhibitors are used to manage chronic pain[9].

Considerable progress has been made towards prevention and control of emesis in cancer therapy. First-generation setrons, including ondansetron, dolasetron, granisetron, and tropisetron, are generally well tolerated by most patients with high antiemetic efficacy. However, the conventional antiemetic regimen is suboptimal for acute-phase and delayed-onset emesis. Palonosetron, the only second-generation setron, addresses some of these shortcomings with a higher binding affinity and a longer half-life[11,12]. Still, refractory emesis and adverse side effects limit clinical use. A better understanding of setrons' antagonistic action on 5-HT$_3$Rs would enhance their therapeutic potential[13,14].

There has been extensive debate on the binding orientation of setrons in 5-HT$_3$R. Previous docking studies have found several energetically favorable poses within the orthosteric binding pocket[15–18]. The predicted orientations of setrons and residue interactions vary significantly among these studies due to uncertainties in homology models and the binding pocket conformation. First high-resolution views of setrons' binding poses came from crystal structures of the acetylcholine-binding protein (AChBP), bound to granisetron, tropisetron, or palonosetron[19–21]. AChBP, a soluble homolog of the pLGIC extracellular domain (ECD), proved to be an excellent surrogate to explain ligand-recognition properties of the pLGIC family. However, the absence of transmembrane and ICDs in AChBP masks the full range of conformational changes exhibited by the channel. These knowledge gaps could be settled by atomic-resolution structures of the full-length 5-HT$_3$R in complex with serotonin and different antagonists. These structures could, in turn, serve as starting points toward the design of more effective therapeutics.

The first structure of the serotonin 3A receptor (5-HT$_3$AR) was solved by X-ray crystallography in the presence of stabilizing nanobodies in an inhibited, non-conducting conformation. Overall, this structure revealed an architecture conserved across the other members of the pLGIC family[22]. Recent high-resolution structures of 5-HT$_3$AR show the channel in an apo conformation and multiple serotonin-bound conformations. The apo conformation was in a closed, resting state, while one serotonin-bound structure was open and the others in either a pre-open or desensitized conformations[23–25]. Together, these structures provide insight into the mechanisms underlying ligand recognition, as well as interdomain conformational coupling that leads to

channel activation and gating. Another recent 5-HT$_3$R structure bound to tropisetron was solved to 4.5 Å showing density for the antagonist in the classic orthosteric site with an occluded pore[24]. The limited resolution of this complex precludes precise orientation of the ligand and surrounding side chains. In addition, the differences between the resting and antagonist-bound states of the 5-HT$_3$AR remain unclear.

In the present study, we have solved the structure of the full-length 5-HT$_3$AR bound to granisetron at 2.92 Å resolution. Granisetron is approved for the prevention of nausea and vomiting, and in combination with corticosteroids, is the recommended regimen for patients undergoing highly emetogenic chemotherapy. We provide details of the granisetron-binding orientation and conformational changes resulting in channel inhibition. In combination with molecular dynamics (MD) simulations and electrophysiology, we have further validated granisetron's binding mode and interactions within the conserved binding pocket.

## Results

**Structural mechanism of 5-HT$_3$AR inhibition by granisetron.** Competitive antagonists bind at the orthosteric site of pLGICs and inhibit channel function by shifting the equilibrium away from the open state (Fig. 1a). This is in contrast to agonists, which bind to the same site but lead to channel opening and eventual desensitization. Granisetron is a competitive antagonist that inhibits serotonin-activated currents at nanomolar concentrations. The effect of granisetron on 5-HT$_3$AR function was measured by two-electrode voltage clamp (TEVC) in *Xenopus* oocyte. Full-length 5-HT$_3$AR messenger RNA (mRNA) was injected into oocytes and currents were recorded 2–5 days post injection. Application of 10 μM serotonin elicited transient currents associated with rapid channel activation and desensitization (Fig. 1b, first pulse). The peak current was significantly inhibited when oocytes were pre-exposed to 1 nM granisetron (Fig. 1b, second pulse). The inhibition was fully reversible when granisetron was washed off and serotonin was subsequently applied (Fig. 1b, third pulse). At 100 nM concentration, granisetron completely eliminates currents induced by 10 μM serotonin (Supplementary Fig. 1).

Single-particle cryogenic electron microscopy (cryo-EM) was used to solve the structure of the full-length 5-HT$_3$AR (sequence shown in Supplementary Fig. 2) in complex with granisetron and infer the molecular details of granisetron inhibition. Detergent-solubilized 5-HT$_3$AR was incubated with 100 μM granisetron for 1 h prior to vitrification on cryo-EM grids. Iterative classifications and refinements produced a final three-dimensional (3D) reconstruction at a nominal resolution of 2.92 Å using a total of 46,757 particles (Supplementary Fig. 3). The local resolution of the map, estimated using ResMap, was in the range of 2.5–3.5 Å (Supplementary Fig. 4). The final map contained density for the entire ECD, transmembrane domain (TMD), and a large region of the intracellular domain (ICD), which were all used in model building and refinement (Supplementary Fig. 5 and Table 1). The overall architecture of granisetron-bound 5-HT$_3$AR (5-HT$_3$AR-granisetron) is similar to previously solved 5-HT$_3$A receptors[22–25]. In the present map, all the domains are well resolved, with the exception of the unstructured MX-MA loop in the ICD (Fig. 1c), making it possible to build a model of the full-length structure (Fig. 1d and Supplementary Fig. 5). The non-protein densities present in the map include three sets of protrusions at the ECD periphery corresponding to N-linked glycans (Fig. 1c, right) and a distinct strong density at each of the intersubunit interfaces in the ECD, corresponding to granisetron (Fig. 1c, e).

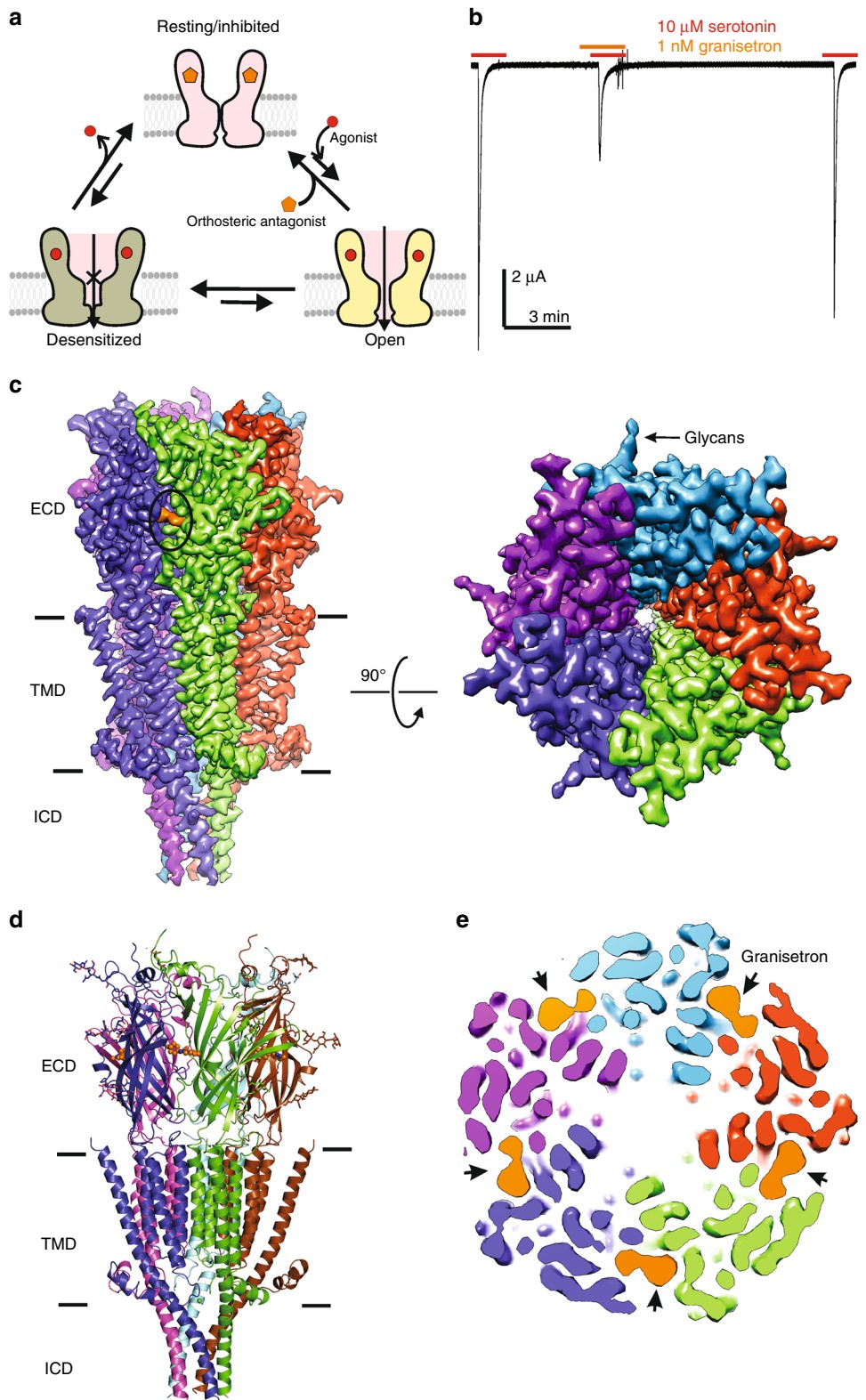

High-quality density maps at and near the ligand-binding site allowed us to model granisetron and its immediate environment, especially the "aromatic cage" formed by surrounding residues at the canonical neurotransmitter-binding pocket, with a certain degree of confidence (Fig. 2a). The stability of the assigned binding pose of granisetron, with the bicyclic ring in a boat/chair conformation and the N-methyl group in axial position in the piperidine chair conformation, was confirmed by MD simulations of the ligand–protein complex in a membrane mimetic environment. This pose ranked more favorably than a binding pose of granisetron with the bicyclic ring in a chair/chair conformation and the N-methyl group in equatorial position in the piperidine chair conformation using metadynamics (discussed later in the text). According to this 5-HT$_{3A}$R-granisetron structure, residues from loops A, B, and C on the principal (+) subunit and loops D, E, and F from the complementary (−) subunit form an enclosure

**Fig. 1** Cryogenic electron microscopy (cryo-EM) structure of granisetron-bound full-length serotonin 3A receptor (5-HT$_{3A}$R). **a** A schematic showing three fundamental conformations that constitute the gating cycle in pentameric ligand-gated ion channel (pLGIC) function: a resting state, a transient open state, and a desensitized state. Agonist-binding shifts the equilibrium towards the open state and then to the high-agonist affinity, desensitized state. Orthosteric (competitive) antagonists exert their effect by shifting the equilibrium towards the resting (or inhibited) state. **b** Trace showing a continuous recording of 5-HT$_{3A}$R currents (−60 mV) in oocytes measured by two-electrode voltage clamp (TEVC) in the presence of serotonin (marked by red line) and pre-applied granisetron (marked by orange line). The effect of granisetron inhibition was fully reversible as seen in the third pulse. **c** Map of full-length 5-HT$_{3A}$R-granisetron reconstructed from 46,757 particles at 2.92 Å resolution. Side-view parallel to the membrane and extracellular view are shown in left and right panels, respectively. Each monomer is shown in a different color for clarity. Density corresponding to granisetron (left panel, circle) and glycans (right panel, arrow) are indicated. **d** Three-dimensional cartoon model of 5-HT$_{3A}$R-granisetron structure generated from EM reconstruction (side view). For each subunit, three sets of glycans are shown as stick representation. **e** Top-view of 5-HT$_{3A}$R-granisetron map sliced at the binding site to show all five granisetron molecules, each bound at the interface of two subunits (indicated by arrows)

| Table 1 Cryo-EM data collection/processing | |
|---|---|
| | **5-HT$_{3A}$-granisetron (EMDB-0469; PDB-6NP0)** |
| Data collection and processing | |
| Magnification | ×130,000 |
| Voltage (kV) | 300 |
| Electron exposure (e⁻/Å²) | 40 |
| Defocus range (μm) | −1.0 to −2.5 |
| Pixel size (Å) | 0.532 |
| Symmetry imposed | C5 |
| Initial particle images (no.) | 243,290 |
| Final particle images (no.) | 46, 757 |
| Map resolution (Å) | 2.92 |
| FSC threshold | 0.143 |
| Refinement | |
| Initial model used (PDB code) | 6BE1 |
| Map sharpening B-factor (Å²) | −50 |
| Model composition | |
| Non-hydrogen atoms | 16,891 |
| Protein residues | 16,300 |
| Ligands | 591 |
| B factors (Å²) | |
| Protein | 111.12 |
| Ligand | 117.85 |
| RMS deviations | |
| Bond lengths (Å) | 0.008 |
| Bond angles (°) | 1.072 |
| Validation | |
| MolProbity score | 1.51 (95th percentile) |
| Clashscore | 3.63 (97th percentile) |
| Poor rotamers (%) | 0.55 |
| Ramachandran plot | |
| Favored (%) | 94.83 |
| Allowed (%) | 5.17 |
| Disallowed (%) | 0.00 |

Cryo-EM cryogenic electron microscopy, FSC Fourier shell coefficient, RMS root mean square, PDB Protein Data Bank code

around granisetron. Residues within 4 Å of granisetron include Trp156 in loop B, Phe199 and Tyr207 in loop C, Trp63 and Arg65 in loop D, and Tyr126 in loop E. These residues are strictly conserved and perturbations at each of these positions impact binding of both granisetron and serotonin[17,26,27]. In comparison to the 5-HT$_{3A}$R-apo and 5-HT$_{3A}$R-serotonin (State 1 structure[25]), these residues appear to have undergone rotameric reorientations (Fig. 2b). However, differences in resolution of the three structures limit a detailed investigation of the changes in side-chain positions in response to binding agonist and antagonists. The cationic center of granisetron is a tertiary ammonium in the bicyclic ring (pKa 9.6) and is positioned at the deep end of the pocket nestled by Trp156 (loop B), Tyr207 (loop C), Trp63 (loop D), and Tyr126 (loop E). The close proximity of the piperidine nitrogen and Trp156 is conducive for a cation–pi interaction, as

predicted from the AChBP-5-HT$_3$ chimera structure[19]. Trp156 is also involved in a similar interaction with the primary amine of serotonin[28]. However, in comparison to serotonin, granisetron is larger in size and extends further outward into the subunit interface such that the aromatic indazole ring is oriented parallel to the membrane and makes a cation–pi interaction with the positively charged nitrogen in the guanidinium group of Arg65 (Fig. 2a). A cation–pi interaction between Arg65 and the indazole ring was predicted from the AChBP-5-HT$_3$ chimera structure[19]. Granisetron's position also requires Arg65 (loop D) and Trp168 (loop F) side chains to reorient (Supplementary Fig. 6a). Interestingly, large orientational differences were predicted for Trp168 when the binding site was occupied by agonist or antagonist[29]. It is noteworthy that mutations at the Arg65 position in human 5-HT$_{3A}$R abolish granisetron binding but tropisetron binding is only reduced[30]. A comparison of 5-HT$_{3A}$R-granisetron and 5-HT$_{3A}$R-tropisetron structures (Supplementary Fig. 6a, b) shows that although the bicyclic rings are in a similar location, the indole/indazole ring are positioned differently[24]. While the model suggests a different side-chain orientation for Arg65 and Trp168 in 5-HT$_{3A}$R-tropisetron, limited resolution in this region prevents unambiguous assignment. Crystal structures of AChBP carrying mutations to mimic the serotonin-binding pocket of 5-HT$_{3A}$R and bound to either granisetron or tropisetron reveal an identical binding pose for the two inhibitors (Supplementary Fig. 6c, d)[19,20]. Overall, the setron binding pose as seen in the 5-HT$_{3A}$R-granisetron structure is distinct from previously reported complexes. Although both granisetron and tropisetron have similar effects on 5-HT$_{3A}$R, tropisetron has been shown to bind to α7nAChR leading to channel activation, while granisetron does not bind to α7nAChR[31,32]. These findings suggest that subtle structural differences between granisetron and tropisetron binding poses may underlie pharmacological variations. Future high-resolution structural information on various 5-HT$_{3A}$R–setron complexes in a native environment will further inform us on the significance of these differences.

As a highly potent competitive antagonist, we expected granisetron to stabilize a conformation similar to the apo-state. Instead, an alignment of 5-HT$_{3A}$R-granisetron with the 5-HT$_{3A}$R-apo structure shows a counter-clockwise twist of β-strands in the ECD, leading to a small inward movement of loop C closing in on granisetron (Figs. 2c, 3a and Supplementary Fig. 7a). The conformation of loop C has been correlated with the ligand occupancy in the binding site and the functional state of the channel. Studies with the AChBP have shown that agonist binding induces a "closure" of loop C, capping the ligand-binding site, a conformational change that may be linked to channel opening in pLGICs. On the other hand, antagonist-bound structures show loop C further extended outward[33]. The 5-HT$_{3A}$R structures solved thus far, in apo- and serotonin-bound states, follow this general trend at the level of loop C[23–25].

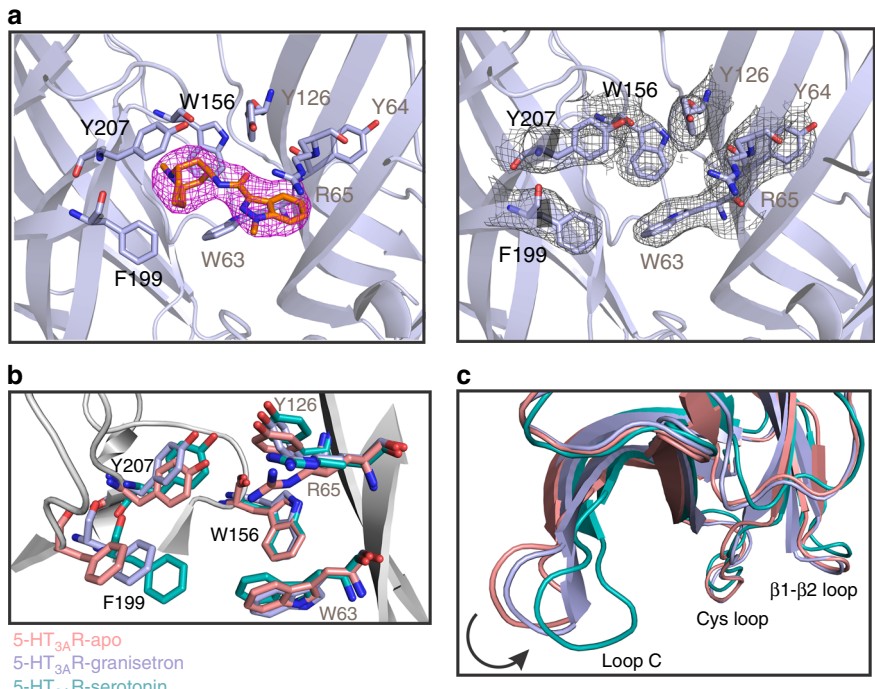

**Fig. 2** The granisetron binding site. **a** The density map of granisetron contoured at 9σ (left) and map around the residues at the binding site located at the intersubunit interface (right). The residue labels on the principal subunit are marked in black and those on the complementary subunit are marked in brown. **b** A comparison of the serotonin 3A receptor-apo (5-HT$_{3A}$R-apo), 5-HT$_{3A}$R-granisetron, and 5-HT$_{3A}$R-serotonin structures shows that residues involved in ligand-binding undergo rotameric reorientation. **c** Alignment of the three structures reveals an inward motion of loop C in 5-HT$_{3A}$R-granisetron relative to 5-HT$_{3A}$R-apo, which is in the direction toward activation as seen in the 5-HT$_{3A}$R-serotonin structure

However, other studies have shown that truncation of loop C does not inhibit unliganded pLGIC gating[34], and hence it is unclear to what extent ligand-induced conformational changes in loop C are related to pore opening.

The loop C conformation in 5-HT$_{3A}$R-granisetron, although open, is distinct from other antagonist loop C conformations because it lies between the open position of loop C in the 5-HT$_{3A}$R-apo structure and the closed position in the 5-HT$_{3A}$R-serotonin structure (Fig. 2c). The TMD of 5-HT$_{3A}$R-granisetron is rotated clockwise compared to 5-HT$_{3A}$R-apo, once again in a position between the apo and serotonin-bound structures (Fig. 3b and Supplementary Fig. 7b). The M2-M3 linker is bent and tilted upwards, but positioned slightly lower than in the 5-HT$_{3A}$R-apo conformation. The modest repositioning of the interfacial loops in the ECD and helices in the TMD results in slightly reduced buried surfaces between adjacent principal and complementary subunits in 5-HT$_{3A}$R-granisetron (3027 Å$^2$) compared to 5-HT$_{3A}$R-apo (3161 Å$^2$). Interestingly, the decrease in buried surface area is reminiscent of recently reported serotonin-bound states, specifically the open-state (3096 Å$^2$) and pre-open or desensitized state (2533 Å$^2$)[25]. Although granisetron induces a rotational movement in the TMD, the pore radius is comparable to 5-HT$_{3A}$R-apo and the permeation pathway presents barriers in all the three domains (Fig. 3c). The first barrier to ion permeation is in the middle of the ECD at Lys108 in the β4–β5 loop at ~2 Å. The M2 helices appear straight with constrictions at 16′, 13′, 9′, 6′, and 2′, which are smaller than the radius of a hydrated Na$^+$[35]. The Leu260 (9′) is located in the middle of the pore and forms the narrowest region of the pore. The intracellular end of M2 is lined by Glu250 which has been implicated in governing charge selectivity[36]. The lateral portals at the interface of the TMD and ICD, which serve as ion exit pathway, are occluded by the post-M3 loop. The periphery regions of the portals are lined by MA/M4, which

appear as a straight helix. Overall, this conformation of the ICD is similar to that seen in the 5-HT$_{3A}$R-apo conformation, and is clearly distinct from the serotonin-bound, open conformation (State 2)[25]. Thus, the 5-HT$_{3A}$R-granisetron structure represents a non-conducting conformation. In agreement, the recently solved tropisetron-bound 5-HT$_{3A}$R[24] shows constrictions at the same locations of the pore. These inhibited states are in contrast to the strychnine-bound GlyR structure, where the pore at Leu277 (9′) is constricted tighter than the radius of a dehydrated chloride ion even though the intracellular end is wide open[37]. However, the recent GABA$_A$R structure bound to bicuculline shows similar constrictions to 5-HT$_{3A}$R-granisetron in the middle and the intracellular end of the pore[38].

Numerous docking studies of granisetron on AChBP and 5-HT$_{3A}$R models have been performed to predict granisetron binding. Because these studies were carried out before 5-HT$_{3A}$R structures were solved, it is clear that many of these predictions were inaccurate with the placement of the indazole ring of granisetron deep in the binding pocket interacting with Trp156 and Tyr207 and the bicyclic ring positioned near Trp63[16,17]. To confirm the stability of the 5-HT$_{3A}$R-granisetron structure, we performed MD simulations with the 5-HT$_{3A}$R-granisetron complex inserted into a phospholipid bilayer mimicking the cell membrane. The overall 5-HT$_{3A}$R pentamer conformation remained close to its density-fitted structure, as evident from small values of the Cα root mean squared deviations (RMSDs) of the secondary structure elements (helices and sheets) of the protein ECD (average RMSD: 1.1 ± 0.1 Å) as well as the entire protein (average RMSD: 2.1 ± 0.2 Å) during 100 ns simulation production run (Fig. 4a, left panel). All five granisetron molecules also remained close to their starting structures as assessed by RMSD of the ligand relative to its cryo-EM inferred pose (Fig. 4a, right panels). To show the type of interactions that these molecules maintain during simulation, we calculated

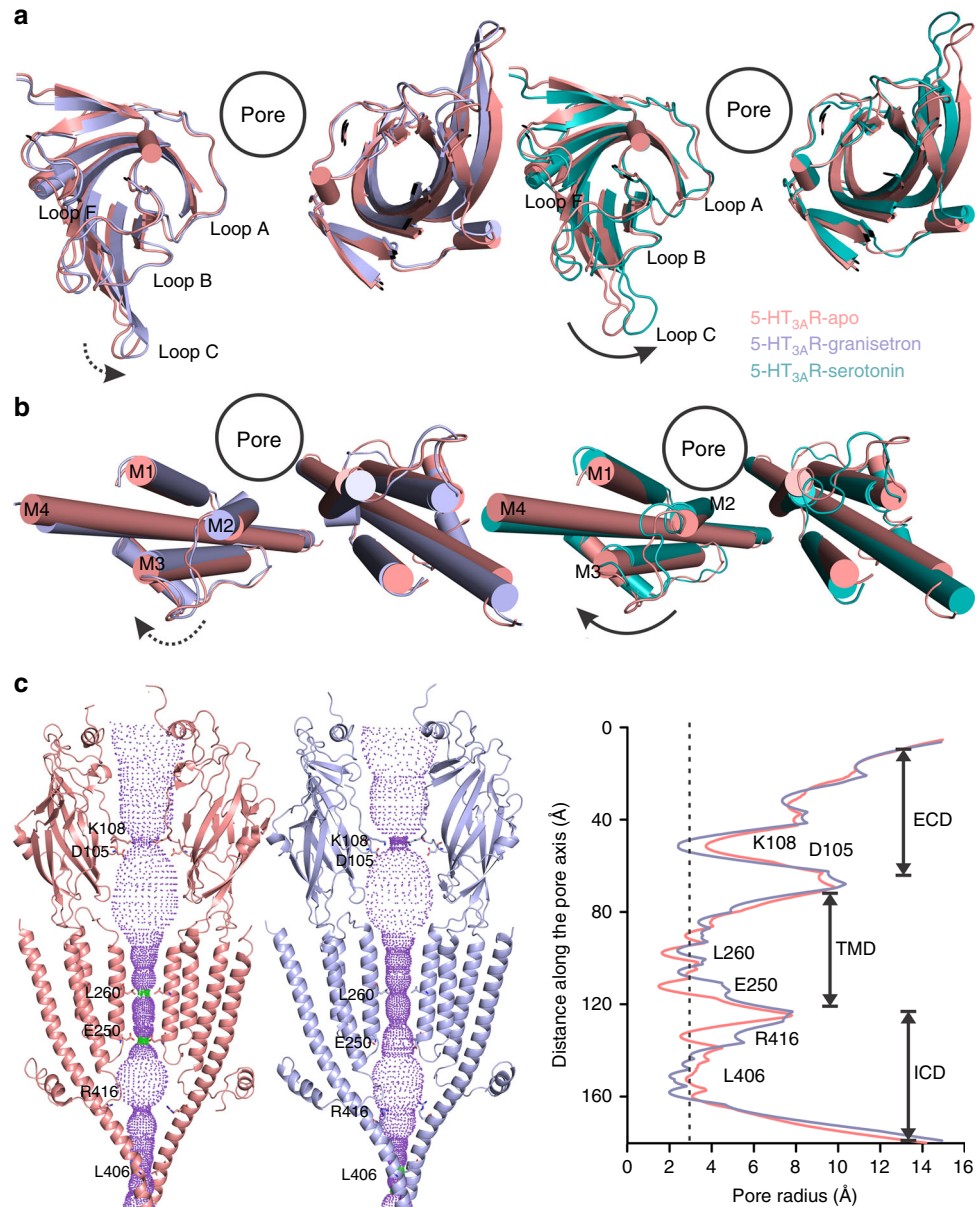

**Fig. 3** Conformational differences between the apo and ligand-bound states. **a** An extracellular view of the extracellular domain (ECD) upon global alignment of serotonin 3A receptor-apo (5-HT$_{3A}$R-apo) structure with 5-HT$_{3A}$R-granisetron (left) and 5-HT$_{3A}$R-serotonin (right). Only ECDs from two non-adjacent subunits is shown for clarity. A counter-clockwise motion of the ECD is observed as indicated by the arrows. The serotonin-induced motion is of larger magnitude compared to that of granisetron, highlighted by the solid and dotted arrows, respectively. **b** A comparison of the transmembrane domains (TMDs) (viewed from the extracellular side) in 5-HT$_{3A}$R-granisetron structure (left) and 5-HT$_{3A}$R-serotonin (right) when aligned with respect to 5-HT$_{3A}$R-apo. Only two non-adjacent TMD subunits are shown for clarity. In both panels, a clockwise rotation of the TMD is observed with 5-HT$_{3A}$R-serotonin revealing a larger change. **c** Pathway of ion permeation of 5-HT$_{3A}$R-apo and 5-HT$_{3A}$R-granisetron generated with HOLE[54]. The cartoon representation of two subunits are shown for clarity. The locations of pore constrictions are shown as sticks. The pore radius is plotted as a function of distance along the pore axis. The dotted line indicates the approximate radius of a hydrated Na$^+$ ion, which is estimated at 2.76 Å (right)[35]

5-HT$_{3A}$R-granisetron interaction fingerprints (see specific interaction types in figure legend) for each protomer in the complex (see results in Supplementary Fig. 8). These fingerprints confirm that binding of each granisetron molecule was mainly stabilized through hydrophobic interactions with 5-HT$_{3A}$R residues and water-mediated polar interactions involving the protonated nitrogen on the piperidine rings with Thr154 and Glu209 only (Supplementary Fig. 8). Since the cryo-EM density could not discriminate between axial or equatorial positions of the N-methyl group in the bicyclic ring of granisetron (Fig. 4b), we used a metadynamics-based strategy[39] to rank the two different

binding poses of the ligand. As shown in Fig.4c, the granisetron pose with the bicyclic ring in boat/chair conformation and the N-methyl group in an axial position in the piperidine chair conformation exhibited a lower average RMSD from its equilibrated pose when compared to the granisetron pose with the bicyclic ring in boat/chair conformation and the N-methyl group in an equatorial position in the piperidine chair conformation, suggesting that the former is a more favorable binding pose.

Extensive mutagenesis studies using natural and unnatural amino acids have been done to probe ligand binding in the 5-HT$_3$

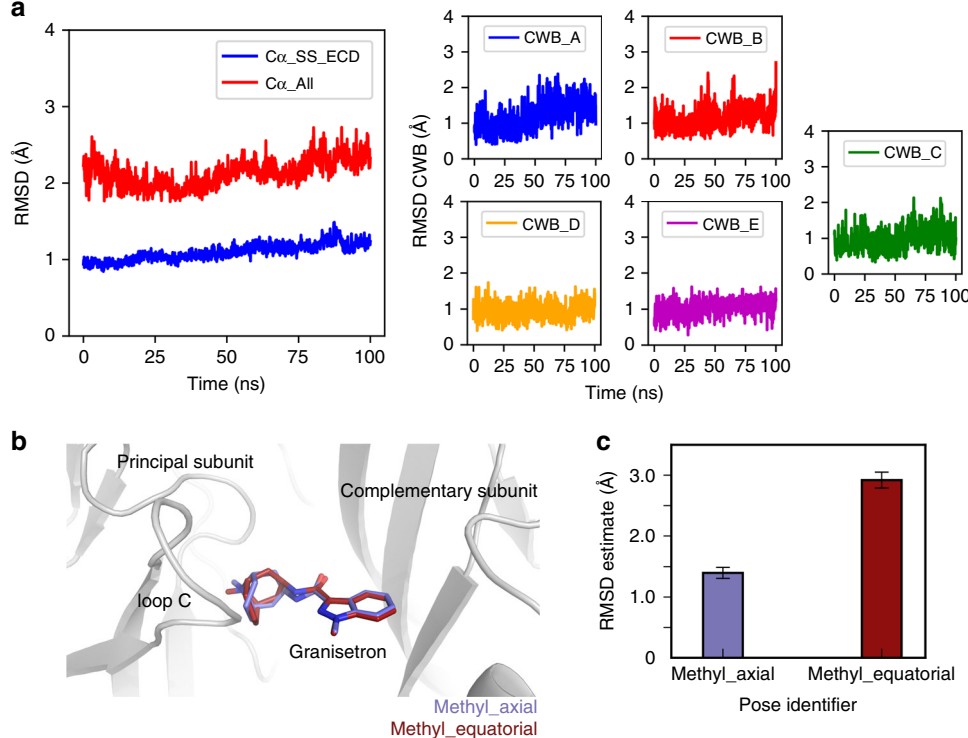

**Fig. 4** Assessment of the overall stability of the granisetron-serotonin 3A receptor (5-HT$_{3A}$R) structure. **a** Time evolution of the root mean squared deviations (RMSD) of Cα atoms of secondary structure elements of the extracellular domain (ECD) and all Cα atoms of the 5-HT$_{3A}$R pentamer (left panel). The RMSD of each of the granisetron molecules (labeled CWB) in each subunit A to E (right panel) calculated with respect to the cryo-EM-derived structure during 100 ns production simulations. **b** Two possible granisetron poses with the bicyclic ring in boat/chair or chair/chair conformation and the N-methyl group in axial or equatorial positions in the piperidine chair conformation. These two poses were used as input for metadynamics-based ranking. **c** Ranking of the two granisetron poses shown in **b** using metadynamics. Error bars represent the standard error of the mean of RMSD estimates from 10 metadynamics simulations. Source data are provided as a Source Data file

receptor[17,18,26–28,30,40]. Radioligand binding studies have revealed the effect of mutational perturbation on the granisetron binding and the reported $K_d$ for [$^3$H]granisetron are wild type (WT) (0.3 nM), W63Y (0.9 nM), R65A (1.8 nM), Y126F (0.9 nM), W156Y (no detectable binding), and Y207F (1.3 nM)[17]. In line with these studies, we find that the extent of granisetron inhibition is significantly reduced when residues in the 5-HT$_{3A}$R binding site, from both the principal subunit (W156Y and Y207F) and the complementary subunit (W63Y, R65A, and Y126F), are mutated (Fig. 5). For competitive antagonists, the extent of inhibition depends on agonist concentration as antagonist effects decrease with increasing agonist concentration. To make a meaningful comparison of granisetron inhibition across various mutations, the serotonin concentration in each case was kept close to the half-maximal effective concentration (EC$_{50}$) value for the mutant. This value was chosen to ensure sufficient current density was evoked in response to serotonin and yet an inhibition was observed with granisetron.

Several studies have supported the notion that antagonists stabilize conformations distinct from canonical resting conformations even though channels remain non-conductive[33,41–43]. In 5-HT$_{3A}$R, serotonin and granisetron interact at the same binding pocket, and interestingly, granisetron induces rotameric reorientations of the side chains in the same direction as serotonin, albeit of smaller magnitude. In addition, the global motion of the granisetron-bound 5-HT$_{3A}$R is in the direction of activation relative to the 5-HT$_{3A}$R-apo structure, but channel activation is not realized in the presence of the antagonist. Binding of granisetron seems to increase the energy barrier required to reach activation thus resulting in the stabilization of an inhibited state.

5-HT$_{3A}$R-granisetron structural model was built from high-resolution cryo-EM map; however, it is to be noted that the structural information derived here comes from detergent-solubilized 5-HT$_{3A}$R in a micelle environment and there may be conformational difference of the antagonist-bound structure in a membrane environment. While future cryo-EM studies in a native environment will address this limitation, results from MD simulations of the 5-HT$_{3A}$R-granisetron structure in a membrane lends confidence to the validity of the proposed binding pose.

Previous 5-HT$_{3A}$R structures have all been homomeric receptors. Incorporation of the B subunit of the heteromeric receptor significantly alters channel activity. Compared to the homomeric channel, the heteromeric 5-HT$_{3AB}$R has a different serotonin dose–response curve, single-channel conductance, permeation properties, desensitization, Ca$^{2+}$ permeability, and pharmacology[12]. However, knowledge of the structural contribution of this subunit to ligand-recognition and inhibition mechanisms is lacking. There is some indication that 5-HT$_{3A}$R may be more abundant in the central nervous system and the 5-HT$_{3AB}$R in the peripheral nervous system. Obtaining structures of heteromeric channels will open the door to answering questions related to ligand occupancy in the binding site and how many ligands must bind for the channel to activate. Our current models cannot answer the question of occupancy because data processing requires averaging of all five ligand-binding sites. Furthermore, a more complete understanding of ligand recognition leading to channel activation and inhibition will require the context of a membrane environment depicting a physiological state. In addition to orthosteric antagonists, strategies targeting allosteric mechanisms in inhibition of

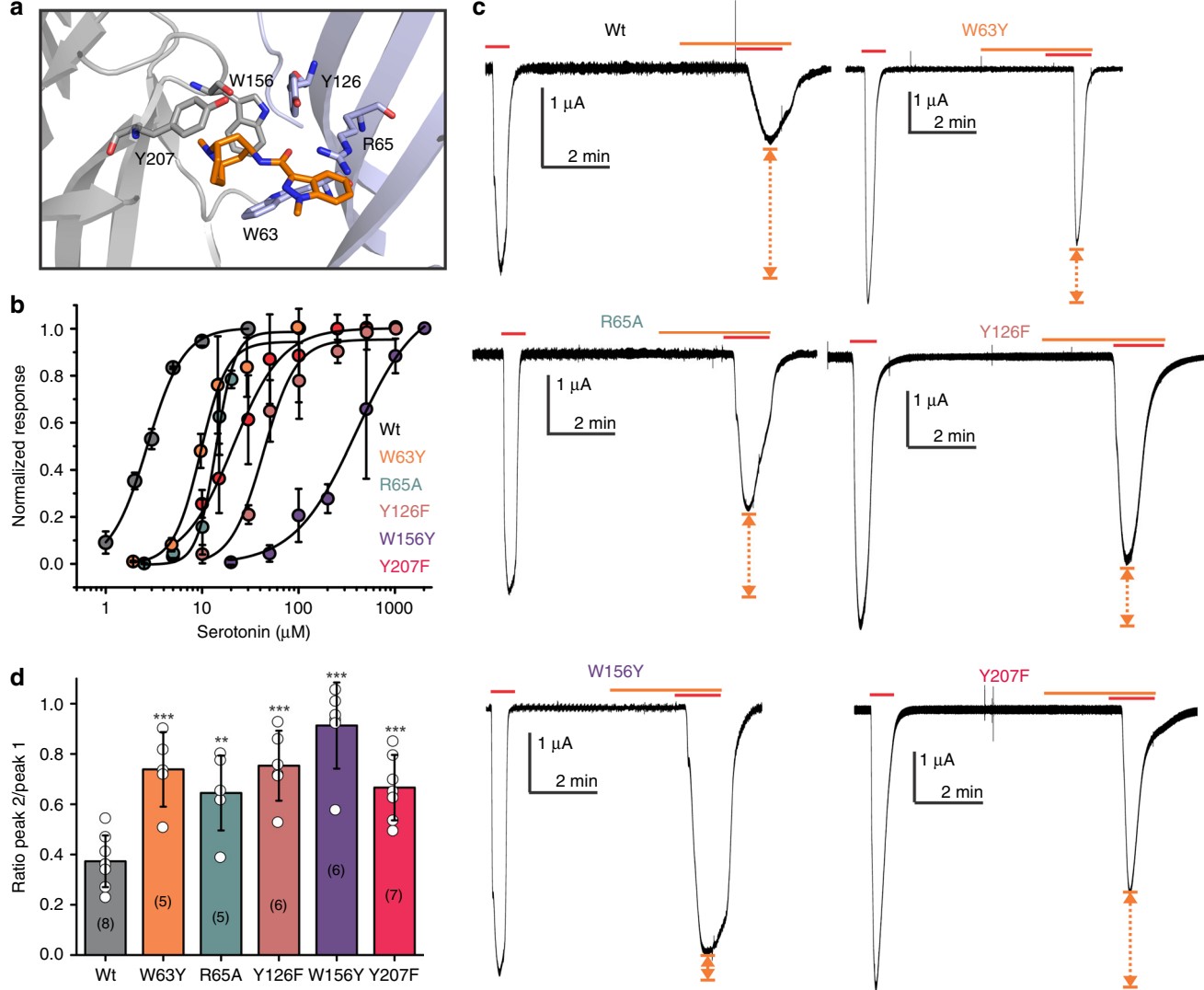

**Fig. 5** Effects of mutations at the ligand-binding pocket on granisetron inhibition. **a** Granisetron interactions with Trp156 and Tyr207 from the principal subunit and Trp63, Arg65, Tyr126 from the complementary subunit are depicted as stick representation. **b** Serotonin dose response measured by two-electrode voltage clamp (TEVC) recordings (at −60 mV) for wild-type (WT) 5-HT$_{3A}$R, W63Y, R65A, Y126F, W156Y, and Y207F mutants, expressed in oocytes. The half-maximal effective concentration (EC$_{50}$), the Hill coefficient (nH), and the number of independent oocyte experiments for WT and mutants are: WT (EC$_{50}$: 2.70 + 0.09 μM; nH: 2.3 + 0.17; n: 3), W63Y (EC$_{50}$: 9.93 + 0.77 μM; nH: 3.1 + 0.72; n: 4), R65A (EC$_{50}$: 13.79 + 0.50 μM; nH: 4.4 + 0.59; n: 4), Y126F (EC$_{50}$: 42.8 + 4.4 μM; nH: 2.6 + 0.71; n: 4), W156Y (EC$_{50}$: 306 + 44 μM; nH: 1.58 + 0.24; n: 4), and Y207F (EC$_{50}$: 20.35 + 1.7 μM; nH: 1.9 + 0.27; n: 5). **c** Currents were elicited in response to serotonin (concentrations used near EC$_{50}$ values of WT and mutants). The following concentrations of serotonin were used: WT-1 μM, W63Y-10 μM, R65A-10 μM; Y126F-40 μM, W156Y-200 μM, and Y207F-20 μM. Currents were measured in response to serotonin (marked by red line) and pre-application of granisetron (marked by orange line). Dotted arrows show the extent of granisetron inhibition. **d** A plot of the ratio of peak current in the presence of granisetron to peak current in the absence of granisetron is shown for WT and mutants. Data are shown as mean ± s.d. (n is indicated within parentheses). Significance at $p = 0.001$ (***) and $p = 0.05$ (**) calculated by two-sample $t$ test for WT and mutants. Source data are provided as a Source Data file

5-HT$_{3A}$R set the stage for the discovery of new drugs that could be receptor subtype specific, and do not alter the pattern of endogenous neurotransmitters. These studies will pave the way for design of drugs that specifically target the 5-HT$_3$R in the central nervous system or gut to treat psychiatric or gastrointestinal-tract-related conditions with less off-target effects.

## Methods

**Electrophysiological measurements in oocytes.** The gene encoding mouse 5-HT$_{3A}$R (purchased from GenScript) was inserted into a *Xenopus laevis* expression vector (pTLN) using cloning primers shown in Supplementary Table 1. DNA linearization was carried out with the *Mlu*1 restriction enzyme overnight at 37 °C. mRNA synthesis was done using the mMessage mMachine Kit (Ambion) as per the

manufacturer's instructions and purification was done with RNAeasy Kit (Qiagen). mRNA for studied mutants was prepared in the same way. Three to ten nanograms of mRNA was injected into *X. laevis* oocytes (stages V–VI) and experiments were performed 2–5 days after injection. For control experiments to verify that no endogenous currents were present, oocytes were injected with the same volume of water. W.F. Boron kindly provided oocytes used in this study. Female *X. laevis* were purchased from Nasco. Animal experimental procedures were approved by Institutional Animal Care and Use Committee (IACUC) of Case Western Reserve University. Oocytes were maintained at 18 °C in OR3 medium (GIBCO-BRL Leibovitz medium containing glutamate, 500U each of penicillin and streptomycin, pH adjusted to 7.5, osmolarity adjusted to 197 mOsm). Two-electrode voltage-clamp experiments were performed on a Warner Instruments Oocyte Clamp OC-725. Currents were sampled and digitized at 500 Hz with a Digidata 1332A. Data were analyzed by Clampfit 10.2 (Molecular Devices). Oocytes were clamped at a holding potential of −60 mV. Solutions were changed using a syringe pump perfusion system flowing at a rate of 6 ml/min. The electrophysiological solutions

consisted of (in mM) 96 NaCl, 2 KCl, 1.8 CaCl$_2$, 1 MgCl$_2$, and 5 HEPES (pH 7.4, osmolarity adjusted to 195 mOsm). Chemical reagents (serotonin hydrochloride, granisetron hydrochloride) were purchased from Sigma-Aldrich. For WT and mutants, the extent of inhibition was assessed by the ratio of peak current in the presence of granisetron over peak current in the absence of granisetron.

**Full-length 5-HT$_{3A}$R cloning and transfection.** Codon-optimized mouse 5-HT$_{3A}$R (NCBI Reference Sequence: NM_001099644.1) was purchased from GenScript (Supplementary Table 2) and inserted into pFastBac1 vector consisting of four strep-tags (WSHPQFEK) at the N terminus, followed by a linker sequence (GGGSGGGSGGGS) and a TEV-cleavage sequence (ENLYFQG) and a C-terminal 1D4-tag[44]. *Spodoptera frugiperda* (Sf9) cells (Expression System) were cultured in ESF921 medium (Expression Systems) without antibiotics and incubated at 28 °C without CO$_2$ exchange. Transfection of sub-confluent cells was done with recombinant 5-HT$_{3A}$R bacmid DNA using Cellfectin II transfection reagent (Invitrogen) according to the manufacturer's instructions. Cell culture supernatants were collected and centrifuged at 72 h post transfection at $1000 \times g$ for 5 min to remove cell debris to obtain progeny 1 (P1) recombinant baculovirus. Sf9 cells infected with P1 virus stock produced P2 virus and subsequently P3 viruses from the P2 virus stock. P3 viruses were used for recombinant protein production.

**5-HT$_{3A}$R expression and purification.** Approximately $2.5 \times 10^6$ per ml Sf9 cells were infected with P3 recombinant viruses. After 72 h post infection, the cells were harvested and centrifuged at $8000 \times g$ for 20 min at 4 °C to separate the supernatant from the pellet. The cell pellet was resuspended in dilution buffer (20 mM Tris-HCl, pH 7.5, 36.5 mM sucrose) supplemented with 1% protease inhibitor cocktail (Sigma-Aldrich). Cells were disrupted by sonication on ice and non-lysed cells were removed by centrifugation ($3000 \times g$ for 15 min). The membrane fraction was separated by ultracentrifugation ($167,000 \times g$ for 1 h) and solubilized with 1% C12E9 in a buffer containing 500 mM NaCl, 50 mM Tris, pH 7.4, 10% glycerol, and 0.5% protease inhibitor by rotating for 1 h at 4 °C. Non-solubilized material was removed by ultracentrifugation ($167,000 \times g$ for 15 min). The supernatant containing 5-HT$_{3A}$ receptors was collected and bound with 1D4 beads pre-equilibrated with 150 mM NaCl, 20 mM HEPES, pH 8.0, and 0.01% C12E9 for 2 h at 4 °C. The beads were then washed with 100 column volumes of 150 mM NaCl, 20 mM HEPES, pH 8.0, and 0.01% C12E9 (buffer A). The protein was then eluted with buffer A supplemented with 3 mg/ml 1D4 peptide (TETSQVAPA)[44]. Eluted protein was concentrated and deglycosylated with PNGase F (NEB) by incubating 5 U of the enzyme per 1 μg of protein for 2 h at 37 °C under gentle agitation. Deglycosylated protein was then applied to a Superose 6 column (GE Healthcare) equilibrated with buffer A. Fractions containing the protein were collected and concentrated to 2–3 mg/ml using 50-kDa MWCO Millipore filters (Amicon) for cryo-EM studies.

**Cryo-EM sample preparation and data acquisition.** 5-HT$_{3A}$R protein (~2.5 mg/ml) was filtered and incubated with 100 μM granisetron for 1 h. 3 mM fluorinated Fos-choline-8 (Anatrace) was added and the sample was incubated until blotting. The sample was blotted twice with 3.5 μl sample each time onto Cu 300 mesh Quantifoil 1.2/1.3 grids (Quantifoil Micro Tools), and immediately the grid was plunge frozen into liquid ethane using a Vitrobot (FEI). The grids were imaged using a 300 kV FEI Titan Krios microscope equipped with a Gatan K2-Summit direct detector camera. Movies containing 30 frames were collected at ×130,000 magnification (set on microscope) in super-resolution mode with a physical pixel size of 0.532 Å/pixel, dose per frame 1.30 e$^-$/Å$^2$. Defocus values of the images ranged from −1.0 to −2.5 μm (input range setting for data collection) as per the automated imaging software EPU.

**Image processing.** MotionCor[45] with a *B*-factor of 150 pixels$^2$ was used to correct beam-induced motion. Super-resolution images were binned ($2 \times 2$) in Fourier space, making a final pixel size of 1.064 Å. All subsequent data processing was conducted in RELION 3.0[46]. The defocus values of the motion-corrected micrographs were estimated using the Gctf software[47]. Approximately 243, 290 auto-picked particles from 1318 micrographs were subjected to 2D classification to remove suboptimal particles. An initial 3D model was generated from the 5-HT$_{3A}$R-apo cryo-EM structure (RCSB Protein Data Bank code (PDB ID): 6BE1). A low-pass filter of 60 Å was applied using EMAN2. Multiple rounds of 3D auto-refinements and 3D classifications generated two good classes with 53,029 and 7072 particles. To further improve the resolution, per-particle motion correction was performed using Bayesian polishing in RELION 3.0[46,48], which was followed by 3D autorefinement and classification, yielding one major class with 46,757 particles. Per-particle contrast transfer function refinement and beam tilt correction were applied followed by a final 3D autorefinement. In the post-processing step in RELION, a soft mask was calculated and applied to the two half-maps before the Fourier shell coefficient (FSC) was calculated resulting in an overall resolution of 2.92 Å (FSC = 0.143 criterion). The *B*-factor estimation and map sharpening were performed in the post-processing step. Local resolutions were estimated using the RESMAP software[49].

**5-HT$_{3A}$R model building.** The map for 5-HT$_{3A}$R-granisetron contained density for the entire ECD, TMD, and a large region of the ICD. The final refined models comprised of residues Thr7–Leu335 and Leu397–Ser462. The missing region (336–396) is of the unstructured MX loop that links the amphipathic MX helix and the MA helix. The 5-HT$_{3A}$R-apo cryo-EM structure (PDB ID: 6BE1) was used as an initial model and aligned to the 5-HT$_{3A}$R-granisetron cryo-EM map calculated with RELION 3.0. Cryo-EM map was converted to the mtz format using mapmask and sfall tools in CCP4i software[50]. The mtz map was then used for manual model building in COOT[51]. After initial model building, the 5-HT$_{3A}$R-granisetron model was refined against its EM-derived map using the phenix.real_space_refinement tool from the PHENIX software package[52], using rigid body, local grid, NCS constraints, and gradient minimization. The model was then subjected to additional rounds of manual model fitting and refinement, resulting in a good final model to map cross-correlation coefficient of 0.809. Stereochemical properties of the model were evaluated by Molprobity[53]. Protein surface area and interfaces were analyzed by using PDBePISA server (http://www.ebi.ac.uk/pdbe/pisa/). To compare the 5-HT$_{3A}$R-apo to 5-HT$_{3A}$R-granisetron all ligands, ions and water molecules were removed from the PDB files. Additional residues in the 5-HT$_{3A}$R-apo structure were also removed before analysis so that surface area comparisons were made between identical construct lengths. The pore profile was calculated using the HOLE program[54]. Figures were prepared using PyMOL v.2.0.4 (Schrödinger, LLC).

**MD simulations.** The cryo-EM-derived structure of the granisetron-bound 5-HT$_{3A}$ pentamer with the bicyclic ring of granisetron in boat/chair conformation and the *N*-methyl group in an axial position in the piperidine chair conformation was prepared for simulations using the default protocol of the Protein Prep Wizard in the Schrodinger suite 2018-3 (Small-Molecule Drug Discovery Suite 2018-3, Schrödinger, LLC, New York, NY, USA). This protocol includes addition of any missing side chains (in this case, Thr8, Ile329, Leu333, and Val399), as well as hydrogen atoms in the starting ligand–protein complex. These steps are followed by side chain protonation assignment, and energy minimization using the OPLS3 force field. The resulting granisetron-bound 5-HT$_{3A}$ pentamer complex was embedded in a 1-palmitoyl-2-oleoyl phosphatidyl choline (POPC) bilayer using the Membrane Builder functionality of the CHARMM-GUI webserver[55]. The system was then solvated with the TIP3P water model plus 0.15 M NaCl, and neutralized with additional sodium ions using the *solvate* and *genion* modules of Gromacs 2018.2[56], respectively. The resulting simulation system had an initial dimension of 136 × 136 × 196 (Å)$^3$ and consisted of the 5-HT$_{3A}$R pentamer, granisetron molecules bound to each 5-HT$_{3A}$R subunit, 450 POPC molecules, ~83,870 water molecules, 242 sodium ions, and 227 chloride ions, for a total of ~345,690 atoms. The CHARMM36m force field[57–59] was used to describe proteins and lipids. Parameters for granisetron were obtained from the ParamChem webserver using the CHARMM general force field[60] (https://cgenff.paramchem.org/). MD simulations were run using Gromacs 2018.2 with a timestep of 4 fs following a steepest descent energy minimization for 5000 steps, as well as 100 ps isothermal–isovolumetric (NVT) and 50 ns isothermal–isobaric (NPT) equilibration runs. The NPT equilibration run was performed in 18 steps, the first 17 of which employed gradually decreasing positional restraints on the heavy atoms of lipids, protein side chains, protein backbone, ligand ring atoms, and lastly, the remaining ligand atoms. The last step consisted of a 2 ns unrestrained NPT equilibration run, which was followed by a 100 ns production run. During equilibration, system temperature and pressure were maintained at 300 K and 1 bar, respectively, using velocity rescale[61] for temperature coupling and a Parrinello–Rahman barostat for pressure coupling. Semi-isotropic pressure coupling was applied during simulations. All bonds involving H-atoms were constrained using the LINCS algorithm[62]. Short-range nonbonded interactions were cut at 12 Å. Long-range electrostatic interactions were computed using the Particle Mesh Ewald summation with a Fourier grid spacing of 1.2 Å. A production run of 100 ns was performed using the same parameters as described above, but switched to the Nose-Hoover thermostat[63]. Trajectory analyses were performed using PyEMMA 2[64] and VMD[65]. Specifically, the RMSD of the protein Cα atoms and granisetron heavy atoms were calculated following superposition of residues of secondary structure elements (helices and sheets) of the protein ECD in each trajectory frame onto the coordinates of the starting cryo-EM structure. Structural interaction fingerprints were calculated using an in-house python script to monitor 5-HT$_{3A}$ interactions with each of the granisetron molecules. Specifically, for each 5-HT$_{3A}$ residue, 5-HT$_{3A}$R-granisetron interactions were calculated as a 9-bit representation based on the following nine types of interactions: apolar (van der Waals) interactions, face-to-face and edge-to-face aromatic interactions, hydrogen-bond interactions with the protein as hydrogen-bond donor or hydrogen-bond acceptor, electrostatic interactions with the protein positively or negatively charged, and one-water or two-water-mediated hydrogen-bond interactions. A distance cutoff of 4.5 Å was used to identify apolar interactions between two non-polar atoms (carbon atoms), while a cutoff of 4 Å was used to identify aromatic and electrostatic interactions. Residue interactions were limited to side chain interactions. The average probabilities and errors for each interaction were estimated using a two-state Markov model, sampling the transition matrix posterior distribution using standard Dirichlet priors for the transition probabilities as described in ref. [66].

**Metadynamics simulations**. A metadynamics-based strategy[39] was used to rank the two granisetron poses that best fit the cryo-EM density and differ for having the bicyclic ring in a boat/chair or chair/chair conformation and the *N*-methyl group in an axial or equatorial position in the piperidine chair conformation. Protein complexes with these two granisetron binding poses were equilibrated for 15 ns using the aforementioned protocol in the "MD simulation" section. After equilibration, 10 metadynamics simulations were run for each binding pose using as collective variable the RMSD of the ligand heavy atoms and the backbone heavy atoms of a stable region of the protein that is far from the binding pocket (residues 209–212 of 5-HT$_{3A}$R subunit A, hereafter referred to as "anchor atoms"). The RMSD was calculated following superposition of the ligand heavy atoms along with the anchor atoms in each simulation frame onto the corresponding atoms in the equilibrated 5-HT$_{3A}$R cryo-EM structure. Metadynamics simulations were run using the PLUMED 2.3.1 plugin[67] within Gromacs 2016.3. Each metadynamics simulation was run for 10 ns with Gaussian hills of height 0.21 kJ/mol and width 0.02 nm deposited every 4 ps. Granisetron binding pose stability was assessed based on the lowest average or thermodynamically most favored RMSD of the ligand heavy atoms plus the anchor atoms from their equilibrated starting positions, calculated after averaging the RMSD estimates obtained from the 10 independent metadynamics runs as follows:

$$\langle s \rangle = \frac{\int \mathrm{d}s\, s\, e^{-(F(s)/k_B T)}}{\int \mathrm{d}s\, e^{-(F(s)/k_B T)}}, \tag{1}$$

where $F(s)$ is the free energy as a function of the collective variable $s$, $k_B$ is the Boltzmann's constant, and $T$ is the temperature.

**Reporting summary**. Further information on research design is available in the Nature Research Reporting Summary linked to this article.

## Data availability

Accession numbers: The coordinates of the 5-HT$_{3A}$R-granisetron structure and the Cryo-EM map have been deposited under PDB ID: 6NP0; EMBD ID: EMD-0469 with the wwPDB and EMDB. All relevant data are available from the corresponding author upon reasonable request. The source data underlying Figs. 4b and 5b, d are provided as a Source Data file.

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

## Acknowledgements

We acknowledge the use of instruments at the Stanford-SLAC Cryo-Electron Microscopy Facility and thank Prof. Wah Chiu for providing us imaging time and supervising the data collection (supported by National Institutes of Health grants: P41GM103832 and S10OD021600). We are grateful to the Cleveland Center for Membrane and Structural Biology for providing the access to the Cryo-EM instrumentation. We thank Denice Major for assistance with hybridoma and cell culture at Department of Ophthalmology and Visual Sciences (supported by the National Institutes of Health Core Grant P30EY11373). We thank Dr. Walter F. Boron for kindly providing us *Xenopus* oocytes and for unrestricted access of the oocyte rig. We are deeply appreciative of the support provided by Dr. Fraser Moss and Mr. Brian Zeise with the oocyte rig. We are very grateful to the members of the Chakrapani lab for critical reading and comments on the manuscript and to Dr. Davide Provasi in the Filizola lab for providing the script that produced the plots in Supplementary Fig. 8. Computations were run on resources available through the Scientific Computing Facility at the Icahn School of Medicine at Mount Sinai and the Extreme Science and Engineering Discovery Environment under MCB080077 (to M.F.), which is supported by National Science Foundation grant number ACI-1548562. This work was supported by the National Institutes of Health grants R01GM108921, R01GM131216, and Cryo-EM supplements: 3R01GM108921-03S1, R01GM108921-5S1 to S.C. and the AHA postdoctoral Fellowship to S.B. (17POST33671152).

## Author contributions

S.B., Y.G. and S.C. conceived the project and designed experimental procedures. S.B. and Y.G. purified the protein and optimized the cryo-EM sample preparation. S.B. carried out the screening, data analysis, model building, and refinement. M.L.M. collected the cryo-EM data. Y.G. performed two-electrode voltage-clamp recordings. A.K. performed the MD simulations, docking, and other computational analyses under the supervision of M.F. S.C. supervised the execution of the experiments, data analysis, and interpretation. S.B., Y.G. and S.C. drafted the manuscript with contributions from A.K. and M.F. All authors reviewed the final manuscript.

## Additional information

**Competing interests:** The authors declare no competing interests.

