## [Peer Review File · Nature Communications]

Reviewers' comments:

Reviewer #1 (Remarks to the Author):

This manuscript describes a new cryo-EM structure of the homopentameric mouse 5-HT3A receptor in complex with the competitive antagonist granisetron. Full-length receptor was solubilized in C12E9 detergent and incubated with granisetron to stabilize the channel conformation in a presumed closed/resting state. The high quality cryo-EM data are of sufficient resolution to model side chains, at least in the ECD, and granisetron, with confidence. The newly resolved structure was compared with the published apo-structure and serotonin-bound structure of the receptor. Docking simulations and electrophysiology on mutants also lend some support the binding mode of granisetron determined by cryo-EM. In addition, the authors describe the different binding modes of two different antagonists, granisetron vs. tropisetron, to suggest how subtle changes in drug chemistry might affect the pharmacological properties. I find this structure to be an important advance- granisetron is an important drug and the EM data are of quite high resolution for the receptor family. In their current presentation, the docking and electrophysiology do not add substantially to what the structure reveals and to what was already known from mutagenesis in the literature. I suggest the impact of this study would be strengthened by considering the following suggestions/comments.

Introduction and abstract:

I think it is an overstatement to say that the mechanisms underlying 'setron' inhibition are poorly understood. There are multiple high resolution structures of a soluble domain bound to tropisetron and granisetron, a full length receptor structure complex with tropisetron, and it has been clear for a long time that these molecules work as competitive antagonists. These structures (here I mainly refer to the AChBP mutant) should be mentioned earlier, and referenced when they are first mentioned. Currently they are mentioned on p. 7 and not referenced until p. 9.

It remains a bit unclear how to reconcile the observed TMD conformation (more open than apo) with the assumed activity of granisetron (inert antagonist). Fig. 1b would be strengthened by adding the response to granisetron alone, given that it stabilizes a conformation in the current structure that appears intermediate between an agonist and apo complex. I suspect the literature is unambiguous about this, but if you were to find that granisetron had very weak partial agonist activity it might make the channel conformation easier to interpret. Or, the absence of lipids means one can't interpret fine details like intermediate states in the TMD.

Thank you for sharing the map and model. They provide high confidence in correct overall modeling of the ligand. I think though that you have some bad bond angles in the ligand- the indazole through carboxamide linkage should be planar, correct? Please check the ligand restraints and geometry.

Chain RMSDs are zero, which suggests that NCS constraints (and not just restraints) were applied. Please mention this, if correct, in the methods.

I also looked at your structure vs. the 2YME pdb. One ligand difference I noticed is in the N-methyl group extending off the bridged nitrogen- it is oriented differently than in the 2YME pdb coordinates from the Ulens lab. I do not know if this difference is meaningful- but, any differences in modeling are a source for discovering errors in the current or previous models. This difference motivated me to superposition a principal subunit from that model onto a principal subunit (chain A on chain A) from your model. In doing so I found that the orientations and in fact geometries of granisetron are quite different- though on page 9 you state that this AChBP mutant structure correctly predicted the pose of granisetron. Your density is very clear- thus while some bond angles need repair, the overall pose must be correct, which is very interesting. I think you missed (in the submitted manuscript) an important opportunity to emphasize the importance of your

structure, as it shows how the soluble LBD model is limited in accuracy of modeling at least this small molecule's binding interactions. Or, if I have misunderstood something, please clarify. A side by side comparison in a figure (or a superposition) would be very helpful whether they are the same or different.

A good place to make comparisons with other models might be in a revised SFig 4. Regardless, in this figure, please be clear here which tropisetron complex you are showing in the existing panels. Please, somewhere, include comparisons with AChBP (mutant) complexes of tropisetron and granisetron as well as the current study's granisetron and the Nury structure of tropisetron. In the AChBP complex, the tropane nitrogen in granisetron was suggested to form a H-bond with the loop B Trp carbonyl- is that interaction supported by the current study? It looks like it is not, and may relate to the differing activities of this class of ligands between nicotinic and serotonin receptors.

Is the resolution of the Nury complex with tropisetron low (as in poor) enough that one really ought not read into that ligand conformation, and local side chains, and one can now assume the AChBP-based orientation was correct for tropisetron? Looking at the density in the Nury paper, the side chain for the R65 is mostly not in density, and the ligand is unclear (and they explicitly say one cannot interpret the ligand conformation with confidence). To me there does not seem to be a strong argument that the detailed comparisons of ligand orientation should be made with the 4.5A tropisetron complex- if you are confident it is correct, great, and say so, but if not, please avoid detailed comparisons, and explain why.

Fig. 3b. The orientation here is confusing. Are we looking from inside or outside the cell? Are those two adjacent or non-adjacent subunits? Maybe depth cueing of the long M4 helix in particular would make it more intuitive.

Also in Fig 3b, both the granisetron-bound structure and serotonin-bound structure slightly move clock-wise compared to apo structure. Then, how different is the granisetron-bound structure from the serotonin-bound structure? A direct comparison would be helpful.

Fig. 3c suggests the ICD portals are occluded- is that the case?

Does the MA-MX region form one continuous helix (looks like it does), and is that consistent with the apo or the 5HT-bound structures published earlier?

The density for the M2-M3 loop is very poor. How was it modeled (what was the basis of modeling it in the current conformation)?

Please discuss the detergent environment. Here C12E9 was used, and no lipids. The Nury structures included PC, PA and CHS. Would that be expected to make the fine details of TMD conformation too speculative to compare?

Why was protein deglycosylated- was that important to get to this resolution? The presence of what look like well-exposed glycans suggest this treatment was not effective in cleaving the sugars.

In Fig. 5 TEVC experiments, one EC50 value for 5-HT was measured in the same experimental setup/lab, and the values for the other two mutants were taken from the literature. The range of published values for EC50s are large, and the changes presented in 5c could seemingly result from not being at the exact same concentration relative to the EC50 for 5-HT for that mutant. I suggest obtaining experimental EC50 values for each mutant, and using those concentrations in this experiment.

How was "model resolution" calculated in Table 1? I think this line should refer to model vs. map FSC or should come from one of the new-ish Phenix comprehensive validation numbers, and

should not be the experimental resolution of the model used to generate the initial reference map in reconstruction (which should be low pass filtered to very low resolution anyway). I confess I find this line in the table confusing/misleading and don't think it's helpful to include.

In Fig 4, CWB_A-CWB_E corresponds to DP1-DP5? It a little bit confusing. Overall, the rationale and presentation in this section is less clear than in the rest of the manuscript. The EM data are so clear in terms of ligand pose. Docking studies should be a mechanism for generating hypotheses, not proving one. How does the discovery here that docking occasionally gives you the right answer merit including it in this study? To me this component weakens the study and for no good reason- the cryo-EM gives you the answer, and then the docking suggests maybe the cryo-EM is not really correct? Another caveat here is that the simulation was done in a POPC bilayer, while the structure was determined in C12E9 detergent- two very different conditions that further hinder comparison of the results.

Reviewer #2 (Remarks to the Author):

The paper overall is interesting and addresses an important question. A great variety of methods are used to elucidate granisetron binding to the ionotropic serotonin receptor. Focusing on the molecular dynamics/ docking study - in my view this part of the manuscript adds important value and links previous docking efforts to the actual experimental binding pose. This is interesting in its own right, and should be common practice following elucidation of bound drugs to target proteins. The approaches used are all valid and convincing. However, the simulations shown in Fig 4 A/B are quite short, especially in 4A a drift in both RMSD curves is still rather evident. I do not think this affects the main conclusion though.

Reviewer #3 (Remarks to the Author):

In two recent papers (Naure Communications 2018, Nature 2018), the Chakrapani group presented cryo-EM structures of apo and full-length serotonin-bound serotonin receptors (5HT3R). This manuscript extends the previous work and presents a new granisetron-bound cryo-EM structures of full-length 5HT3-Rs resolved at 2.92 angstrom. The new structure reveals granistrone, a competitive antagonist, binding pose and highlights its interactions with 5HT3R sidechains. The authors compare the new structure to the apo- and agonist-structures to provide some insights into competitive antagonist inhibition mechanisms. The manuscript is clear and well written. Approach valid, quality of data good, MD simulations and mutagenesis combined with electrophysiology experiments add some support for the conclusions. I am somewhat on the fence about the impact of this work for Nature Communications. A recent 5-HT3Rs structure bound to tropisetron (a related competitive antagonist) was recently published in Nature (2018), albeit at lower resolution 4.5 angstroms. In addition, most of the insights from the new structure about binding pose and mechanism of granisetron action are not unexpected based on years of structure-activity experiments and previous crystal structures of an engineered soluble serotonin soluble binding protein (EMBO reports, 2012). However, one could argue that having a high-resolution structure provides important confirmation. It is important to note that a cryo-EM structural model from detergent solubilized receptors solved in detergent micelles may not represent the native antagonist bound structure.

One weakness in the manuscript is that the functional data are limited. The mutations of key residues had somewhat small effects on the ability of granisetron to inhibit serotonin currents. Only one concentration of granisetron was assayed. It is unknown how the mutations alter the IC50 values for granisetron inhibition of serotonin currents. One would expect if these residues were key for high affinity binding of granisetron that greater than 10-fold changes in IC50 values would be seen. For W156Y, the current decay appears to lag after removal of serotonin + granisetron, which

is a bit surprising.

As pointed out by authors, due to resolution of tropisetron structure (Nature 2018), it is difficult to assign its orientation. Thus, their detailed comparisons between tropisetron orientation in the binding site and granistrone orientation should be tempered and may not be warranted (supplemental figure 4). In addition, one needs to be careful claiming significant differences in rotameric orientations of binding site residues when comparing between apo structure (4.3 angstrom resolution), serotonin-bound structure (3.3 angstrom) and the granistrone bound structure (2.9 angstrom). Due to different resolutions of the structures, one needs to be careful claiming that changes in rotameric conformations of side chains observed are significant. Whether these differences are seen in a native channel and have a functional impact on receptor function is not known.

Page 7. Reference needed for the AChBP-5-HT3 chimera structure. Reference needed for statement "Trp 156 is also involved in a similar interaction with the primary amine of serotonin". Also, need to reference that cation- π interaction between Arg 65 and indazole ring of granistrone was previously predicted from AChBP-5-HT3 chimera structure. Provide readers a position for W168 (e.g. what binding loop it is located on, Loop F).

Page 8. Previous work from the Auerbach lab has shown that much of Loop C can be removed in the nAChR and the receptor can still gate. Thus, conformation of Loop C may not be an absolute predictor of the physiological conformation state of the receptor.

In summary, the structure provides a high-resolution picture of the orientation of granistrone bound to the 5HT3R. Whether small motions in ECD predicted from comparing apo versus granistrone-bound structures are part of a mechanism underlying granistrone's ability to inhibit channel opening is, at this point, mainly speculative.

Minor points:

It would be helpful for the general reader in the field for the authors to include a complete sequence of the 5HT3R that was used for cryo-EM and is seen in the structures (supplementary figure). Secondary structures and important binding site regions should be labeled. Residues important for granistrone binding highlighted.

Page 19, What missing side chains and hydrogen atoms in the granisetron-bound 5-HT3AR cryo-EM structure were added?

Figure 4c. Is the stability of the docking pose that is similar to the cryo-EM pose significantly different than any of the other poses?

Figure 5 b – please include a time scale bar.

Supplemental Figure 5, CWB_E panel – list the residues in the same order as the other panels for easier comparison. There is a very limited description and discussion of the results of this figure in the paper (one sentence on page 10). The authors may want to expand on their description of these data and summarize any new insights from the data.

Reviewers' comments:

We thank all three Reviewers for their time to review our manuscript. We are pleased with the positive feedback and thankful for the in-depth evaluation of the manuscript and for the insightful comments. We have incorporated all the suggested changes. We hope the Reviewers will find the revision satisfactory and the manuscript acceptable for publication.

Reviewer #1 (Remarks to the Author):

This manuscript describes a new cryo-EM structure of the homopentameric mouse 5-HT3A receptor in complex with the competitive antagonist granisetron. Full-length receptor was solubilized in C12E9 detergent and incubated with granisetron to stabilize the channel conformation in a presumed closed/resting state. The high quality cryo-EM data are of sufficient resolution to model side chains, at least in the ECD, and granisetron, with confidence. The newly resolved structure was compared with the published apo-structure and serotonin-bound structure of the receptor. Docking simulations and electrophysiology on mutants also lend some support the binding mode of granisetron determined by cryo-EM. In addition, the authors describe the different binding modes of two different antagonists, granisetron vs. tropisetron, to suggest how subtle changes in drug chemistry might affect the pharmacological properties. I find this structure to be an important advance- granisetron is an important drug and the EM data are of quite high resolution for the receptor family.

In their current presentation, the docking and electrophysiology do not add substantially to what the structure reveals and to what was already known from mutagenesis in the literature. I suggest the impact of this study would be strengthened by considering the following suggestions/comments.

We thank the Reviewer for the positive feedback and comments. We appreciate that the Reviewer had taken the time to look carefully at the PDB and the .mrc file to evaluate our model. We have incorporated all the suggestions and have included new data to further strengthen the paper. In particular, this revised submission accounts for the following changes:

- 1. Granisetron molecule was corrected for planarity. The stability of two closely related granisteron poses that cannot be distinguished by cryo-EM density was assessed by metadynamics and 100 ns molecular dynamics simulations of the ligand-protein complex in a membrane mimetic environment.*
- 2. Docking results were removed and the stability assessment was limited to the two closely related binding modes of granisetron based on the cryo-EM density.*
- 3. Electrophysiology data were added for two additional mutants and EC50 values were determined for all the tested mutants.*
- 4. The text and figures were edited per suggestion.*

Introduction and abstract:

I think it is an overstatement to say that the mechanisms underlying ‘setron’ inhibition are poorly understood. There are multiple high resolution structures of a soluble domain bound to tropisetron and granisetron, a full length receptor structure complex with tropisetron, and it has been clear for a long time that these molecules work as competitive antagonists. These structures (here I mainly refer to the AChBP mutant) should be mentioned earlier, and referenced when they are first mentioned. Currently they are mentioned on p. 7 and not referenced until p. 9.

We have changed the statement to “the molecular mechanisms underlying setron binding and inhibition of 5-HT_{3A}R are not fully understood”

Included in the introduction now is the discussion of AChBP-setron structures. “There has been extensive debate on the binding orientation of setrons in 5-HT_{3R}. Previous docking studies have found several energetically favorable poses within the orthosteric binding pocket¹⁵⁻¹⁸. The predicted orientations of setrons and residue interactions vary significantly among these studies due to uncertainties in homology models and the binding pocket conformation. First high-resolution views of setrons-binding poses come from crystal structures of the acetylcholine binding protein (AChBP), bound to granisetron, tropisetron, or palonosetron¹⁹⁻²¹. AChBP, a soluble homologue of the pLGIC extracellular domain, proved to be an excellent surrogate to explain ligand-recognition properties of the pLGIC. However, the absence of transmembrane and intracellular domains prevents the protein from achieving the full spectrum of conformational changes exhibited by the channel. These knowledge gaps could be settled by atomic-resolution structures of the 5-HT_{3R} binding site in complex with serotonin and different antagonists. These structures could, in turn, serve as starting points toward the design of more effective therapeutics.”

It remains a bit unclear how to reconcile the observed TMD conformation (more open than apo) with the assumed activity of granisetron (inert antagonist). Fig. 1b would be strengthened by adding the response to granisetron alone, given that it stabilizes a conformation in the current structure that appears intermediate between an agonist and apo complex. I suspect the literature is unambiguous about this, but if you were to find that granisetron had very weak partial agonist activity it might make the channel conformation easier to interpret. Or, the absence of lipids means one can’t interpret fine details like intermediate states in the TMD.

Granisetron does not evoke current on its own and does not act as a partial agonist. As mentioned by the Reviewer, its property as a competitive antagonist is well established in the literature. Per suggestion, we have included a new Supplemental Figure 1 to show that granisetron does not evoke currents even up to 100 nM and at this concentration it completely eliminates activation by 10 μM serotonin. The 5-HT_{3A}R-grani structure is unambiguously a non-conductive conformation. However, the interesting feature of this new structure is that the conformation of this inhibited state is slightly different from the resting state. Further, the competitive inhibitor rather than evoking conformational changes opposite to that of the agonist, seems to partly stabilize a change in the direction of activation, albeit not to the complete extent.

Thank you for sharing the map and model. They provide high confidence in correct overall modeling of the ligand. I think though that you have some bad bond angles in the ligand- the indazole through carboxamide linkage should be planar, correct? Please check the ligand restraints and geometry.

Thank you for pointing it out. We have fixed this and uploaded the new PDB on the rcsb webpage (new validation report included)

Chain RMSDs are zero, which suggests that NCS constraints (and not just restraints) were applied. Please mention this, if correct, in the methods.

Yes, it is now mentioned in the methods.

I also looked at your structure vs. the 2YME pdb. One ligand difference I noticed is in the N-methyl group extending off the bridged nitrogen- it is oriented differently than in the 2YME pdb coordinates from the Ulens lab. I do not know if this difference is meaningful- but, any differences in modeling are a source for discovering errors in the current or previous models. This difference motivated me to superposition a principal subunit from that model onto a principal subunit (chain A on chain A) from your model. In doing so I found that the orientations and in fact geometries of granisetron are quite different- though on page 9 you state that this AChBP mutant structure correctly predicted the pose of granisetron. Your density is very clear- thus while some bond angles need repair, the overall pose must be correct, which is very interesting. I think you missed (in the submitted manuscript) an important opportunity to emphasize the importance of your structure, as it shows how the soluble LBD model is limited in accuracy of modeling at least this small molecule's binding interactions. Or, if I have misunderstood something, please clarify. A side by side comparison in a figure (or a superposition) would be very helpful whether they are the same or different.

The Reviewer is right, there are differences in the orientation and geometry of granisetron molecule in the AChBP (2YME) and 5-HT_{3A}R structures. Our original comment was meant to indicate that the Ulens paper correctly predicted that the bicyclic ring is positioned within the principal subunit and the indazole ring extended toward the complementary subunit. We have removed this line and discuss the differences in the binding pose as seen in the two structures. New Supplemental Figure 6 includes a side-by-side comparison of the granisetron and tropisetron binding poses from 5-HT_{3A}R and AChBP structures.

A good place to make comparisons with other models might be in a revised SFig 4. Regardless, in this figure, please be clear here which tropisetron complex you are showing in the existing panels. Please, somewhere, include comparisons with AChBP (mutant) complexes of tropisetron and granisetron as well as the current study's granisetron and the Nury structure of tropisetron. In the AChBP complex, the tropane nitrogen in granisetron was suggested to form a H-bond with the loop B Trp carbonyl- is that interaction supported

by the current study? It looks like it is not, and may relate to the differing activities of this class of ligands between nicotinic and serotonin receptors.

Per suggestion, we have modified Supplemental Figure 6 to include published structures of setrons complexes with 5-HT_{3A}R and AChBP. The Reviewer rightly notes that the piperidine nitrogen in the bicyclic ring does not form H-bond interaction with Trp156 (loop B) as seen in the AChBP structure. The N-O distance is 3.0 Å in AChBP-granisetrone compared to 4.6 Å in the 5-HT_{3A}R-granisetrone structure. Since it is not unexpected that the ligand may move within the binding-pocket, this interaction may still be feasible.

Interestingly, the H atom on the piperidine nitrogen of the bicyclic ring keeps pointing away from Trp156 during the 100 ns MD simulations of the deposited cryo-EM structure of the 5-HT_{3A}R-granisetrone complex with all granisetrone bicyclic rings in a boat/chair conformation and all N-methyl groups in an axial position in the chair conformation. In contrast, simulations of the complex with 4 out of 5 granisetrone poses with the bicyclic ring in a chair/chair conformation and the N-methyl group in an equatorial position in the same chair conformation reveals stable formation of a hydrogen bond for only 3 out of the 4 molecules during the simulated timescale (data not shown in the manuscript but available upon request).

Is the resolution of the Nury complex with tropisetron low (as in poor) enough that one really ought not read into that ligand conformation, and local side chains, and one can now assume the AChBP-based orientation was correct for tropisetron? Looking at the density in the Nury paper, the side chain for the R65 is mostly not in density, and the ligand is unclear (and they explicitly say one cannot interpret the ligand conformation with confidence). To me there does not seem to be a strong argument that the detailed comparisons of ligand orientation should be made with the 4.5Å tropisetron complex- if you are confident it is correct, great, and say so, but if not, please avoid detailed comparisons, and explain why.

Agreed. The density for R65 in the 5-HT_{3A}R-tropisetron is poor. We have toned down the discussion pertaining to the comparison of the poses and their potential significance.

Fig. 3b. The orientation here is confusing. Are we looking from inside or outside the cell? Are those two adjacent or non-adjacent subunits? Maybe depth cueing of the long M4 helix in particular would make it more intuitive.

Our apologies for the lack of clarity in the figure. We have redone this figure and with details of the view in the figure legend.

Also in Fig 3b, both the granisetrone-bound structure and serotonin-bound structure slightly move clock-wise compared to apo structure. Then, how different is the granisetrone-bound structure from the serotonin-bound structure? A direct comparison would be helpful.

We have now included an alignment of all the three structures (Supplemental Figure 7)

Fig. 3c suggests the ICD portals are occluded- is that the case?

Yes, this is correct. We now state this explicitly in the text.

Does the MA-MX region form one continuous helix (looks like it does), and is that consistent with the apo or the 5HT-bound structures published earlier?

Yes, the MA-M4 region is one continuous helix in the 5-HT_{3A}R-apo, 5-HT_{3A}R-granisetron and one of the 5-HT_{3A}R-serotonin conformations (non-conducting, State 1). Only in the fully-open 5-HT_{3A}R-serotonin conformation (State 2), this region appears as two separate helices.

The density for the M2-M3 loop is very poor. How was it modeled (what was the basis of modeling it in the current conformation)?

Continuous density for the M2-M3 linker is now shown in the Supplemental Figure 5.

Please discuss the detergent environment. Here C12E9 was used, and no lipids. The Nury structures included PC, PA and CHS. Would that be expected to make the fine details of TMD conformation too speculative to compare?

Nury et al structures were also in C12E9 detergent, although their purification protocols included lipid mixtures. There were no major differences in the TMD conformation of the two structures.

Why was protein deglycosylated- was that important to get to this resolution? The presence of what look like well-exposed glycans suggest this treatment was not effective in cleaving the sugars.

Yes, this is a good point. It is likely that this step is unnecessary. Since the protein behaved well, we haven't looked into the sample without the PNGase treatment.

In Fig. 5 TEVC experiments, one EC50 value for 5-HT was measured in the same experimental setup/lab, and the values for the other two mutants were taken from the literature. The range of published values for EC50s are large, and the changes presented in 5c could seemingly result from not being at the exact same concentration relative to the EC50 for 5-HT for that mutant. I suggest obtaining experimental EC50 values for each mutant, and using those concentrations in this experiment.

Per suggestion, we have now measured the EC50 values for all of the tested mutants. This data is included in Figure 5.

How was “model resolution” calculated in Table 1? I think this line should refer to model vs. map FSC or should come from one of the new-ish Phenix comprehensive validation numbers, and should not be the experimental resolution of the model used to generate the initial reference map in reconstruction (which should be low pass filtered to very low resolution anyway). I confess I find this line in the table confusing/misleading and don't think it's helpful to include.

Agreed. We have removed this information from the table.

In Fig 4, CWB_A-CWB_E corresponds to DP1-DP5? It a little bit confusing. Overall, the rationale and presentation in this section is less clear than in the rest of the manuscript. The EM data are so clear in terms of ligand pose. Docking studies should be a mechanism for generating hypotheses, not proving one. How does the discovery here that docking occasionally gives you the right answer merit including it in this study? To me this component weakens the study and for no good reason- the cryo-EM gives you the answer, and then the docking suggests maybe the cryo-EM is not really correct? Another caveat here is that the simulation was done in a POPC bilayer, while the structure was determined in C12E9 detergent- two very different conditions that further hinder comparison of the results.

In Fig 4, CWB_A-CWB_E do not correspond to the DP1-DP5 docking poses but rather to the RMSD time evolution of the different granisetron molecules bound at the 5 different binding pockets of the 5HT3A pentamer, starting from the granisetron binding pose that best matched the electron density and was also found to be the most stable pose as per metadynamics-based ranking. Docking and metadynamics simulations were originally carried out to verify that the granisetron binding pose proposed to best fit the cryo-EM density is not only one of the energetically preferred poses, but also the most stable one.

Although reasonably high, a 2.9 Å resolution is not sufficient to determine unambiguously a small-molecule binding conformation, and computational methods have been used to validate the pose assignment. Our docking coupled with metadynamics did not suggest that the cryo-EM is not correct but rather provided further support to the cryo-EM-inferred pose, demonstrating its higher stability in the binding pocket compared to others. To avoid confusion, we eliminated the docking presentation from this revised manuscript and limited the metadynamics-based assessment of pose stability to the two granisetron poses that best fit the electron density but cannot be discriminated based on it. Although many structures are determined in detergent, the purpose of a simulation in a lipid mimetic such as a POPC bilayer is to verify that the solved structure is stable in a more realistic cell mimetic environment.

Reviewer #2 (Remarks to the Author):

The paper overall is interesting and addresses an important question. A great variety of

methods are used to elucidate granisetron binding to the ionotropic serotonin receptor. Focusing on the molecular dynamics/ docking study - in my view this part of the manuscript adds important value and links previous docking efforts to the actual experimental binding pose. This is interesting in its own right, and should be common practice following elucidation of bound drugs to target proteins. The approaches used are all valid and convincing. However, the simulations shown in Fig 4 A/B are quite short, especially in 4A a drift in both RMSD curves is still rather evident. I do not think this affects the main conclusion though.

We thank the Reviewer for noting the importance of the MD study in this work. As also recognized by the Reviewer, longer simulations would not affect conclusions. We demonstrated this by doubling the length of simulations and submitting the corresponding results in this revised manuscript. Both former and extended new simulations were run for the sole purpose of verifying the overall stability of the identified structural minimum in a lipid mimetic. Although a slight drift can still be seen in the RMSD of C α atoms of secondary structural elements of the ECD, differences from the initial cryo-EM structure are very small (RMSD < 1.5Å). (Please see Figure 4 and Supplemental Figure 8).

Reviewer #3 (Remarks to the Author):

In two recent papers (Naure Communications 2018, Nature 2018), the Chakrapani group presented cryo-EM structures of apo and full-length serotonin-bound serotonin receptors (5HT3R). This manuscript extends the previous work and presents a new ganisetron-bound cryo-EM structures of full-length 5HT3-Rs resolved at 2.92 angstrom. The new structure reveals ganistrans², a competitive antagonist, binding pose and highlights its interactions with 5HT3R sidechains. The authors compare the new structure to the apo- and agonist-structures to provide some insights into competitive antagonist inhibition mechanisms. The manuscript is clear and well written. Approach valid, quality of data good, MD simulations and mutagenesis combined with electrophysiology experiments add some support for the conclusions. I am somewhat on the fence about the impact of this work for Nature Communications. A recent 5-HT3Rs structure bound to tropisetron (a related competitive antagonist) was recently published in Nature (2018), albeit at lower resolution 4.5 angstroms. In addition, most of the insights from the new structure about binding pose and mechanism of ganisetron action are not unexpected based on years of structure-activity experiments and previous crystal structures of an engineered soluble serotonin soluble binding protein (EMBO reports, 2012). However, one could argue that having a high-resolution structure provides important confirmation. It is important to note that a cryo-EM structural model from detergent solubilized receptors solved in detergent micelles may not represent the native antagonist bound structure.

We agree with the reviewer's point that the cryo-EM structural model, from detergent solubilized receptors solved in detergent micelles, may not truly represent the native antagonist bound structure. While future cryo-EM studies in a more native membrane condition will address this limitation, we have presented here MD simulations of the structure in a membrane mimetic environment. The finding that the drug-binding pose deposited with the cryo-EM structure is significantly more stable than another one also fitting reasonably well the electron density lends further confidence to the model. (Noted also in the text, discussion).

One weakness in the manuscript is that the functional data are limited. The mutations of key residues had somewhat small effects on the ability of granisetron to inhibit serotonin currents. Only one concentration of granisetron was assayed. It is unknown how the mutations alter the IC₅₀ values for granisetron inhibition of serotonin currents. One would expect if these residues were key for high affinity binding of granisetron that greater than 10-fold changes in IC₅₀ values would be seen. For W156Y, the current decay appears to lag after removal of serotonin + granisetron, which is a bit surprising.

We acknowledge the weakness cited by this Reviewer. We have addressed this concern by presenting EC₅₀ values for all the mutants tested. To compare the granisetron effect across mutants, we studied the inhibition due to 1 μ M granisetron at serotonin concentrations near the EC₅₀ value for the mutant. For each of the mutants tested in this study, the K_d values for granisetron, measured by radioligand binding studies, have been previously reported. We now provide these values in the text with the references. Consistent with previous studies, each of these mutations alter granisetron inhibition with the most prominent effect seen for W156Y. The binding-site residues have been extensively studied and we provide these references in the text.

As pointed out by authors, due to resolution of tropisetron structure (Nature 2018), it is difficult to assign its orientation. Thus, their detailed comparisons between tropisetron orientation in the binding site and granisetron orientation should be tempered and may not be warranted (supplemental figure 4). In addition, one needs to be careful claiming significant differences in rotameric orientations of binding site residues when comparing between apo structure (4.3 angstrom resolution), serotonin-bound structure (3.3 angstrom) and the granisetron bound structure (2.9 angstrom). Due to different resolutions of the structures, one needs to be careful claiming that changes in rotameric conformations of side chains observed are significant. Whether these differences are seen in a native channel and have a functional impact on receptor function is not known.

We agree with the Reviewer. In response to the comments on the resolution of the previously published tropisetron structure, we have toned down the detailed comparison. Supplemental Figure 6 now includes various setron-binding poses as seen in 5-HT_{3A}R and AChBP structures to broadly highlight the positional difference of our new structure compared to the others (Also requested by Reviewer 1).

We add a cautionary note regarding interpreting the differences in the rotameric orientations of the binding-site residues in the apo, serotonin-, and granisetron- bound structures. “In comparison to the 5-HT_{3A}R-apo and 5-HT_{3A}R-serotonin (State 1 structure²²), these residues appear to have undergone rotameric reorientations (Fig. 2b). However, differences in resolution of the three structures limit a detailed investigation of the changes in side-chain positions in response to binding agonist and antagonists.”

Page 7. Reference needed for the AChBP-5-HT3 chimera structure. Reference needed for statement “Trp 156 is also involved in a similar interaction with the primary amine of serotonin”. Also, need to reference that cation- π interaction between Arg 65 and indazole ring of granistrone was previously predicted from AChBP-5-HT3 chimera structure. Provide readers a position for W168 (e.g. what binding loop it is located on, Loop F).

Done

Page 8. Previous work from the Auerbach lab has shown that much of Loop C can be removed in the nAChR and the receptor can still gate. Thus, conformation of Loop C may not be an absolute predictor of the physiological conformation state of the receptor.

Finding from the Auerbach group shows that deletion of loop C does not inhibit unliganded gating. Therefore they conclude that the agonist-induced closing motion of loop C may not be a requirement for channel opening. However, this study does not argue against the relationship between the direction of loop C motion and the occupancy of the ligand in the binding pocket. We modified the text as follows.

“Studies with the AChBP have shown that agonist binding induces a “closure” of loop C, capping the ligand-binding site, a conformational change that may be linked to channel opening in pLGIC. On the other hand, antagonist-bound structures show loop C further extended outward³³. The 5-HT_{3A}R structures solved thus far, in apo and serotonin-bound states, follow this general trend at the level of loop C²³⁻²⁵. However, other studies have shown that truncation of loop C does not inhibit unliganded pLGIC gating³⁴, and hence it is unclear to what extent ligand-induced conformational changes in loop C are related to pore opening.”

In summary, the structure provides a high-resolution picture of the orientation of granistrone bound to the 5HT3R. Whether small motions in ECD predicted from comparing apo versus granistrone-bound structures are part of a mechanism underlying granistrone’s ability to inhibit channel opening is, at this point, mainly speculative.

Minor points:

It would be helpful for the general reader in the field for the authors to include a complete sequence of the 5HT₃R that was used for cryo-EM and is seen in the structures (supplementary figure). Secondary structures and important binding site regions should be labeled. Residues important for granisetron binding highlighted.

We present this information in the new Supplemental Figure 2.

Page 19, What missing side chains and hydrogen atoms in the granisetron-bound 5-HT₃AR cryo-EM structure were added?

Added on page 18. The missing side chain atoms were those of residues Thr8, Ile329, Leu333, and Val399) and the hydrogen atoms were those of the entire granisetron-bound 5-HT₃AR cryo-EM structure.

Figure 4c. Is the stability of the docking pose that is similar to the cryo-EM pose significantly different than any of the other poses?

Yes, the metadynamics strategy ranked one of the cryo-EM poses better than other top-ranked poses suggested by automated docking. However, in response to reviewer #1's critique, these docking studies have been removed from this revised manuscript.

Figure 5 b – please include a time scale bar.

Done

Supplemental Figure 5, CWB_E panel – list the residues in the same order as the other panels for easier comparison. There is a very limited description and discussion of the results of this figure in the paper (one sentence on page 10). The authors may want to expand on their description of these data and summarize any new insights from the data.

In this revised manuscript, a new representation of the fingerprint analysis is provided alongside a slightly expanded description (page 11).

REVIEWERS' COMMENTS:

Reviewer #1 (Remarks to the Author):

I am satisfied with the authors' response and do not need to see another revision.